# The World Is Not a Theorem

**DOI:** 10.3390/e23111467

**Published:** 2021-11-06

**Authors:** Stuart Kauffman, Andrea Roli

**Affiliations:** 1Institute for Systems Biology, Seattle, WA 98109, USA; stukauffman@gmail.com; 2Department of Computer Science and Engineering, Campus of Cesena, Alma Mater Studiorum Università di Bologna, I-47522 Cesena, Italy; 3European Centre for Living Technology, I-30123 Venezia, Italy

**Keywords:** diachronic evolution of the biosphere, affordance, set theory, incompleteness

## Abstract

The evolution of the biosphere unfolds as a luxuriant generative process of new living forms and functions. Organisms adapt to their environment, exploit novel opportunities that are created in this continuous blooming dynamics. Affordances play a fundamental role in the evolution of the biosphere, for organisms can exploit them for new morphological and behavioral adaptations achieved by heritable variations and selection. This way, the opportunities offered by affordances are then actualized as ever novel adaptations. In this paper, we maintain that affordances elude a formalization that relies on set theory: we argue that it is not possible to apply set theory to affordances; therefore, we cannot devise a set-based mathematical theory to deduce the diachronic evolution of the biosphere.

## Prologue

*[L’universo] è scritto in lingua matematica, e i caratteri son triangoli, cerchi, ed altre figure geometriche, senza i quali mezi è impossibile a intenderne umanamente parola; senza questi è un aggirarsi vanamente per un oscuro laberinto.* (“The universe is written in the language of mathematics, and its characters are triangles, circles, and other geometrical figures, without which it is humanly impossible to understand a single word of it; without these, one is wandering around in a dark labyrinth.” Galileo Galilei, *Il Saggiatore*, Roma, Italy, 1623. English translation by S. Drake).

## 1. Introduction

Egyptian papyri dating back some 5000 years document the use of arithmetic and geometric notions for solving practical problems, such as the need to measure and subdivide the soil [1]. After thousands of years, three centuries before Christ, we find the most remarkable development of these notions: the *Elements*, by the Greek Euclid of Alexandria, a prominent and influential deductive theory—queen of the Sciences and true of the world: the language in which the universe is written is made of triangles and circles, as Galileo stated in the XVII century. Nevertheless, an axiomatic and deductive system such as the *Elements* does not need to describe the world, as shown by non-Euclidian geometries. However, mathematical systems shine as pure crystals and one might expect them to be consistent and complete, as indeed Hilbert did. These hopes have been destroyed by Gödel, who has set the limits of deductive systems. We remark here that these deductive systems are purely syntactic, without semantic referents. Despite these limits, we currently rely on mathematical models for understanding systems of any sort, conscious of the possible incompleteness of the deductions we make. However, there is a kind of incompleteness we have probably overlooked: what can be entailed by a formal system is already contained in it, and we cannot expect to be able to deduce novelty, to entail the becoming of the biosphere.

In this paper, we maintain that the flourishing evolution of the universe cannot be captured by a mathematical model based on set theory. Of course, we can model biological organisms that have appeared during evolution in a given moment of time (*synchronic* modeling), but we contend that this is not possible for *diachronic* evolution modeling.

In Section 2, we start by discussing the notion of *affordance*, which is central in the becoming of the biosphere. Section 3 illustrates the core of our thesis: affordances defeat any sound definition of “set”; therefore, as evolution is enabled by affordances, no mathematical model based on set theory can be used to model the evolution of the biosphere. Several objections might be raised to argue against our claim: in Section 4, we analyze and answer to the ones we think are the most relevant. Finally, in Section 5, we observe that the essence of our statement can be already found in the late writings by Wittgenstein and we summarize the main points of this paper, emphasizing that, rather than projecting a dark and negative perspective, our claim has a creative and positive potential.

## 2. Affordances and the Becoming of the Biosphere

The notion of *affordance* was introduced by Gibson [2] with the aim of expressing the actions that an object enables for an animal observing it. The concept has been subsequently extended, and it is currently adopted in diverse fields such as biosemiotics [3], cybernetics and robotics [4]. In general and abstract terms, we can say that an affordance is “The use of *X* to accomplish *Y*”, where *X* may be, e.g., an object or a living being (as an affordance can be a feature of another organism), and *Y* is in general an action, a behavior, or a biological function. Organisms not only exploit affordances in the environment, but they also mutually create affordances among themselves [5,6]. When the mutual affordances are seized by heritable variation and natural selection, they come to exist. Therefore, we here focus on *morphological variations*. Heritable variations and selection make it possible for organisms to find and incorporate into their structure, function or behavior new, advantageous uses afforded by objects or other organisms. By *organism*, we mean a *Kantian Whole*: an organized self-constructing being, having the property that *the parts exist for and by means of the whole* [7,8]. Self-constructing Kantian Wholes are examples of autopoietic systems [9]. In evolution, affordances open possibilities that organisms can seize by means of heritable variations and selection, such as the case of Darwinian preadaptations or Gould’s exaptations. A paradigmatic example is the emergence of the swim bladder, which is believed to have evolved from the lungs of lung fish. Water got into a lung, now containing both air and water, and afforded a new possible use: The swim bladder affords a means to measure neutral buoyancy in the water column. A new function has come to exist in the evolving biosphere [10]. In general, features of the world come to exist mutually and reciprocally. Further notable examples are flight feathers. These evolved earlier for thermoregulation but were co-opted for the new function of flight. Lens crystallins originated as enzymes [11].

Biosemiotics emphasizes the fact that an affordance is an entangled property of both the organism and its environment. In other terms, affordances cannot be defined in a non-circular way. Affordances are referential degrees of freedom, not independent features of the world. Niches are a prominent example of this, as they constitute the world of an organism. Niche and the organism living in it cannot be defined non-circularly [12].

Montévil nicely explains this aspect discussing the notion of novelty [13]: as for the books in the *Library of Babel* by Borges [14], where all the 410 pages-long books composed of all the possible combinations of characters are stored, what distinguishes a book from a printed random sequence of letters is the *semantic meaning* that the reader gives to the sequence of letters; the meaning depends upon the experience, the history of the reader. In addition, the experience of the reader in turn depends upon the environment in which they live. Again, affordances cannot be defined non-circularly.

Kantian Wholes can contain smaller Kantian Wholes. A bacterium or Archaeon is a Kantian Whole. A eukaryotic cell is a Kantian Whole containing smaller Kantian Wholes comprised of the mitochondria and chloroplasts in the eukaryotic cell. A multi-celled organism is itself a still higher-level Kantian Whole containing eukaryotic cells containing mitochondria and chloroplasts. Communication occurs up and down this hierarchy and at each level. The mixed microbial communities living in the gut of animals are Kantian Wholes reciprocally adapted and constitute reciprocal niches for one another and their host [15]. Each species is the condition for its own existence and that of the others.

Critically, in the course of evolution, these Kantian Wholes evolve ever new morphological and behavioral functions at their diverse levels by Darwinian preadaptation co-opting existing molecules and structures to incorporate new affordances for new uses and functions.

In the physical sciences within the Newtonian Paradigm [16], the phase space of the system is prestated and fixed. For example, consider a billiard table with fixed boundaries, the edges of the table. These boundary conditions fix all possible positions and momenta of the balls on the table. *This set of prefigured possibilities constitutes the phase space of the system*. Then, given laws of motion in differential equation form and initial conditions, integration of the equations yields the entailed trajectories of the system in its prefigured phase space. Quantum mechanics remains in the Newtonian Paradigm. Integration of Schrödinger’s equation yields an entailed propagation of a probability distribution. Von Neumann’s measurement projection process then breaks determinism to yield the Actual true or false outcomes via the Born rule.

The diachronic evolution of the biosphere escapes the Newtonian Paradigm. The functions and behaviors of parts and of organisms are relevant features that constitute ever-changing aspects of the ever-changing relevant phase space of the evolving biosphere. We cannot say ahead of time what these new functions will be. The capacity of an elephant to cool itself using its trunk to shower itself on a hot day is relevant to its survival. Laws of motion cannot be written for phase spaces that change in ways that cannot be prestated. Therefore, there are no entailing laws for the becoming of the biosphere [10,17,18].

In physics, given laws of motion and a prefigured phase space, the evolution of the system is an entailed deduction. Cells literally construct themselves [19]. Thus, the evolution of the biosphere is a non-entailed propagating construction. Construction is not deduction.

## 3. Set Theory and the Limits of Mathematical Modeling of Affordances

In this section, we show that affordances and the diachronic evolution of the biosphere—and all the systems with analogous properties, such as technological evolution—are inherently not subject to a mathematization in terms of set theory. We first observe that the universe is not ergodic, non-ergodic, above the level of atoms [10,18,20]. For example, the universe will not make all possible proteins with 200 amino acids. There are $20^{200}$ possible proteins of this length. This is $10^{260}$. The age of the universe is $10^{17}$ seconds, the shortest time scale is the Planck time scale of $10^{-43}$ seconds. There are about $10^{80}$ particles. If all these were making 200-amino-acid-long proteins in parallel, each Planck time moment it would take $10^{37}$ times the age of our universe to make all these possible proteins just once [10,18].

Most complex things will never get to exist. However, the human heart exists in the universe. How? The simple answer is that life emerged, organisms are Kantian Wholes sustained by their parts, and hearts evolved to sustain their Kantian Whole by pumping blood. Organisms with hearts evolve and carried with them their sustaining and ever-evolving hearts [20]. The example of the elephant as a whole organism cooling itself by showering itself with cold water on a hot day demonstrates that the behavior of an entire organism is functionally relevant to its survival.

Critically, the function of a part is that subset of its causal features that sustains the whole. The function of the heart is to pump blood, not make heart sounds. We cannot explain the existence of hearts in the universe without the concepts of evolving life, Kantian Wholes, and heritable variation and natural selection. Selection is a downward cause, acting on the Kantian Whole thus selecting its sustaining parts [8,10]. That which is ever-selected is sustaining old functions and seizing new affordances that are incorporated as new adaptations. In contrast to the claims of S. Weinberg [21], in evolving biospheres the explanatory arrows really point upward to selection on the Kantian Whole.

One might argue that if we had a computational model for the evolution of the biosphere, in principle we could run the model and observe the realization of some evolutions, among all the possible ones. Non-ergodicity would not be an issue, but just a factor introducing computational limitation, as only a tiny fraction of possible futures can be computed. Nevertheless, such a computational model would require to specify sets of affordances, which is not possible, as we are going to show.

### We Cannot Use Set Theory with Respect to the Emergence of Affordances

We begin with two points:(*i*)Affordances are “The Possible use of X to do Y”.(*ii*)Any physical object has diverse subsets of causal features each of which can constitute an affordance.

A typical example of affordance in this context is that of the uses of an engine block. Obviously, it can be used as propulsive component in a car, it is used both as engine and chassis for tractors, it can be used to crack open coconuts on one of its corner; its cylinder bores can host bottles of wine, the engine block can be used also as a paper weight, and so on. Similarly, Caledonian crows learn to use sticks in a diversity of ways to accomplish a diversity of tasks.

It is important to our considerations that there is no ordering relation between the different uses of an engine block, which are merely a nominal scale. Moreover, there is no deductive procedure that can deduce a new use given the previous observed uses of an object [8,10,18]. Given that we are using an engine block as a superb, if excessive, paper weight, we cannot deduce from this use that the corners of the engine block can be used to crack open coconuts. These points are equally true for proteins in living cells. The same given protein may conduct electrons, absorb photons, bind ligands, catalyze a reaction, serve as a strut in the cytoplasm, carry a tension load or provide a structure on which molecular motors can walk.

Consider the screwdriver today in London. What are some uses? Screw in a screw. Open a can of paint. Wedge a door closed. Scrape putty off the window. Break the window. Scratch my back. Tie to a stick and spear a fish. Rent the spear for 5% of the catch. Lean against a wall, lean plywood held up by the screwdriver and shelter a wet oil painting from the rain…

How many uses of a screwdriver alone or with other things are there? Is the number of uses a specific number? 17? 213? No. Is the number infinite? How would we know? The universe really is non-ergodic. We could not have used a screwdriver to short an electric connection in the year 1265 A.D. We can do so now. The number of uses of a screwdriver alone or with other things is *indefinite*. We have no idea now what new uses of a screwdriver may turn up in the next 1000 years.

In analogy with intuitionistic mathematics, according to which real numbers are processes that develop in time, instead of being fully determined entities [22], we observe that we cannot provide a general description of the affordances of an object before constructing them. In sum, the number of uses of *X* is indefinite in the non-ergodic universe, the uses are unordered, and the different uses of the same object cannot be deduced from one another [8,10,18].

The astonishing implication is that we cannot use Set Theory with respect to the diachronic emergence of new affordances. A first axiom of set theory is the *axiom of extensionality*: “Two sets are identical if and only they contain the same members” [23,24]. However, we cannot prove that the uses of a screwdriver are identical to the uses of an engine block. We cannot prove, once and for all, the uses of *X*! Therefore, there is no axiom of extensionality.

One way to define numbers uses set theory. The number “0” is defined as the set of all sets each of which has 0 elements [23]. In our case, this corresponds to “the set of all objects that have exactly 0 uses.” Well, this is not true. Thus, there are no numbers by this use of set theory. The alternative approach to numbers is via Peano’s Axioms [25]. These require a null set and a successor relation. However, we have no null set. More, the different uses of *X* are unordered. We have no successor relation.

Again, astonishingly, with respect to diachronically emerging morphological and behavioral adaptations via seizing affordances, there are no numbers; no integers, no rational numbers, no equations such as 2 + 3 = 5. There are no equations. Therefore, there are no irrational numbers; no real lines; no equations with variables; no imaginary numbers; no quaternions; no octonions; no Cartesian spaces; no vector spaces; no Hilbert spaces; no union and intersection between the uses of X and Y; no first order logic; no combinatorics; no topology; no manifolds; no differential equations on manifolds.

Worse, the *axiom of choice* [26], which is introduced whenever a choice function cannot be defined, cannot be applied. The axiom of choice is equivalent to “well ordering”, i.e., the axiom of choice is equivalent to the assertion that every set is well-orderable (see Theorem 14.4.3 in [27]), but an ordering among the (diachronic) uses of *X* cannot exist. Without an Axiom of Choice, we cannot take limits, as both the (ϵ−δ) formal definition of limits [28] and the one based on *infinitesimals* [29] rely on Set Theory.

Can we obtain the Axiom of Choice in analogy to real numbers, for which the axiom of choice and set theory hold? No: We observe that real numbers may be given as non-repeating sequences of symbols from a *predefined and finite alphabet*. We know the symbols composing these infinite strings. In the case of new affordances coming to exist over time, such as the uses of the engine block alone or with other things, the uses have not yet come to existence; therefore, the syntactic names have not come to existence, so we do not know the syntactic symbols we would use to name the emerging new uses of an engine block.

We are pointing to a major issue. In the evolution of the biosphere, *semantics*, the functional use of *X* to do *Y*, *precedes any syntactic symbols* we may later use in a mathematical model [5]. In short, “possible uses of X” are affordances seized by heritable variation and natural selection and *become semantic adaptive features* of evolving Kantian Wholes by which they literally construct themselves and *thereby get to exist* for a while in the non-ergodic universe. The *semantic* meaning of the adaptation is “I get to exist”.

Mathematics propagates only syntactic symbols devoid of semantics. No new information can be generated by theorems that is not already in the axioms and rules of inference from the axioms. Shannon’s predefined information in syntactic bits can only be transmitted from the source which already has the information without or perhaps with loss. No new information can be generated. The emergence of ever new morphological and behavioral adaptations in the propagating construction of the biosphere is precisely the generation of ever-new semantic information.

An immediate implication of our results is that animal behavior, including the brain, cannot be identical to a non-embodied Turing system. The reason is simple: Turing systems are purely syntactic. The behaviors of organisms are semantic doings in the world. Syntax arises later as a “named encoding and categorization” of behaviors. The names are taken to be definite, *true* or *false*. The behaviors in the world are not definite.

Unlike physics that is taken to be an entailed flow from a fundamental theory with a prestated Hilbert space of all the possibilities [17], there are *no axioms for the evolution of the biosphere, and no external source of information* for that evolution. We suggest that the ever new-in-the-universe possibilities that emerge as organisms construct themselves and get to exist for some time constitute the creation of new information. It is this propagating construction of new possibilities that is seized and breathes fire into the evolving biosphere. (“Even if there is only one possible unified theory, it is just a set of rules and equations. What is it that breathes fire into the equations and makes a universe for them to describe? The usual approach of science of constructing a mathematical model cannot answer the questions of why there should be a universe for the model to describe. Why does the universe go to all the bother of existing?” [30]).

We claim that any theory requiring the notion of sets that tries to model the diachronic emergence of affordances is inherently flawed [5,8,10,31].

## 4. Possible Objections

We understand that our claim might sound either trivial or provoking, or both. In the following we anticipate what we believe are the main objections that may possibly be raised against our argument.

Objection

If we could find a suitable level of description of the universe, then evolution processes through affordances could be modeled as trajectories in this huge space. The relevant trajectories would be extremely rare in this space, as some specific given real numbers are among reals; this set might even have measure zero, but still exist.

Rebuttal

The assumption that it is possible to find a suitable level of description of the universe means that we are able to define a suitable phase space, and implies that we have identified the relevant observables, symmetries and laws for completely describing the universe. These efforts have worked with spectacular success in particle physics and General Relativity, both of which are confirmed to the 13th decimal place. Here the explanatory arrows really do point downward [21], meaning that the reasons to explain a high-level behavior are to be found at the lower level. In evolving biospheres, the explanatory arrows really do point *upward* to that which comes to exist, i.e., downward causation is in place and, therefore, it is the higher level that we have to observe to explain the behavior at the lower level. There can be no trajectory even of zero measure in any theory in which some of the explanatory arrows point upward.

It is of course possible to provide an explanation a posteriori of an affordance that has emerged, as we can explain heart once we observe and recognize it in an organism. This is not to foretell the coming into existence of the heart.

Objection

Physicists often use mathematics to devise *effective theories* of the physical world and are well aware of their limitations. Are there such *effective theories of the evolution of the biosphere* such that some features of the evolution of the biosphere can still be entailed, even if some approximations will be introduced such that there will be some discrepancies between expected and actual results?

Rebuttal

In the case of emerging affordances, *we do not know the sample space of the process* for the reasons stated above; therefore, *we cannot define a probability measure*, nor can we define what is *random*, so we *cannot estimate our error* [17,32]. Our efforts to model the diachronic evolution of the biosphere are not approximations, *they are simply not applicable*. We do not know the still not existent variables that will become relevant. Elephants really survive, in part, by cooling via spraying cool water with their evolved trunks. Not only do we not know what *will* happen, we do not even know what *can* happen [10,18].

More formally, we can view evolution as an asynchronous parallel branching process: each single step represents the actualization of a possibility offered by an affordance. Since we cannot define the set of all the possible affordances in a given context, it is not possible to state a distribution probability over the possible branches of a node. As a consequence, any prediction of such a model would be just a guess—possibly informed. We emphasize that this kind of failure of prediction is of different nature than the one in chaotic dynamical systems, where we can estimate an error.

Objection

This strong claim is just an overstatement or a misinterpretation of Laplace’s statement concerning an intellect “which at a certain moment would know all forces that set nature in motion, and all positions of all items of which nature is composed, if this intellect were also vast enough to submit these data to analysis, it would embrace in a single formula the movements of the greatest bodies of the universe and those of the tiniest atom; for such an intellect nothing would be uncertain and the future just like the past would be present before its eyes.” (Laplace, P.S. *A Philosophical Essay on Probabilities*, translated into English, ed. by Truscott, F.W. and Emory, F.L., Dover Publications, New York, 1951).

Rebuttal

First of all, we are clearly referring to human impossibility. Moreover, we are not just saying that we, as scientists, have only access to incomplete information on “forces” and “positions of all items of which nature is composed” and we have limited computational capabilities; therefore, our deductions are affected by some error. We are claiming that we cannot even know the functional items that are relevant for our predictions.

## 5. Conclusions

We are often pervaded by a profound sense of wonder when we observe the flourishing and creative power of the biosphere, and we are often also amazed by the way animals and humans find creative solutions to problems or invent and build new tools. These phenomena are enabled by affordances, which also constantly change and appear. For more than twenty-five centuries, we have used mathematics to model phenomena of interest, and so provide explanations to understand what we observe or to control systems and processes. Some intrinsic limits of mathematical systems have been already raised in the Twentieth century, but mathematical models are still successfully applied in a plethora of cases. We would, therefore, expect it to be possible to formalize the evolution of the biosphere, to formally model the diachronic emergence of affordances. Nevertheless, we face here an inherent limit: the properties of circularity and so self-reference, and non-ergodicity characterizing affordances defeat any sound applicability of set theory, and consequently all mathematics depending on it. If our arguments are correct, we can create no mathematical model of the diachronic evolution of the biosphere based on set theory. Perhaps new kinds of mathematics can be invented. In the light of our conclusion, some notable attempts to provide a suitable mathematical framework for evolution and self-referentiality deserve more attention and should be studied in more depth [33,34,35,36,37]. However, this is out of the scope of this paper, which is focused on the limits of set theory.

In rather astonished summary, we claim that the morphological and behavioral evolution of the biosphere, besides the impossibility of being entailed [10,17,18] is inherently not mathematizable in terms of sets. The World Is Not A Theorem. Apologies to Pythagoras, Plato, Neoplatonists, Newton, Bohr, and…

Starting from a discussion on the meaning of words in a language and moving to the foundation of mathematics, Wittgenstein had to some extent already contended that mathematical inferences do not bring new knowledge. In *Remarks on the Foundation of Mathematics* [38], he indeed maintains that surprise cannot come from an inference—if any surprise should come, it is simply because something was not yet understood.

We are aware that our claim might be considered as a negative result that brings shadow and further limits the range of mathematical understanding of the reality. On the contrary, we believe that this incompleteness is a way to be creative [18] and achieve a deeper awareness of our membership in an ever-creative world: “We are of Nature, not above Nature” [8].

## Epilogue

*However, how many kinds of sentence are there? Say assertion, question and command? There are countless kinds; countless different kinds of use of all the things we call “signs”, “words”, “sentences”. Furthermore, this diversity is not something fixed, given once for all; but new types of language, new language-games, as we may say, come into existence, and others become obsolete and get forgotten. We can get a rough picture of this from the changes in mathematics.* (Wittgenstein, Philosophical investigations, first. publ. in 1953, 4th ed. 2009, Wiley-Blackwell, UK).

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
