# Peer review of "The World Is Not a Theorem"

_entropy, 2021, doi:10.3390/e23111467_

Round 1

Reviewer 1 Report

I have read with great pleasure, and great honour the article by Prof. Kauffman and Prof. Roli; The world is not a theorem.

I think the take home message of the article is clear and will be of great interest not just to Entropy readers, but scientist of all domains in general. Thus I accept the article for publication. However, I would like the authors to comment on certain points that I see critical to be openly discussed in an open peer-review process. It is not necessary this discussion be moved to the article per-se, although I leave it to the authors' freedom to incorporate the elements that came convenient from this discussion. What I think is that for the Entropy readers it is important and interesting to have a view of the starting point from which the authors place their arguments.

1-   The Hilbert formalization programme, more than ‘deductive’, refers to the attempt to make the universe of mathematics become a purely syntactic universe, where there are no semantic referents. In this universe, which is computable and unambiguous, therefore self-referential (semantic) systems (the sort of Gödel incompleteness theorems, or Russell vicious circles) cannot exist ‘effectively’, the so-called Church-Turing thesis. The Number Theory and its impossible formalization is a clear example that the universe of mathematics goes beyond syntax and enters the semantic field. Thus, the authors instead of talking about "deductive mathematical systems", which can cause confusion, they should talk about "syntactic mathematical systems", from which one cannot deduce the full emerging of meanings that semantic (self-referential) systems can display. In other words, just as biological organization is self-referential, in the self-constructive sense, it is impossible to predefine the sample space of the process or define a probability measure of it with a syntactic mathematical system. Therefore, the problem is not in the deduction per se, but in the attempt to, first, reduce the universe of mathematics to syntax alone excluding semantic referents (the Hilbert formalization programme), and second, to do not consider a bigger universe of mathematical systems that can deal with semantic systems, e.g., category theory (Robert Rosen’s M,R-system), or calculus of indication (Spencer Brown’s Laws of form).

2-   I understand that Prof. Kauffman has chosen to call organisms self-constructing Kantian Wholes with good reason. However, I would like to know why the authors instead to call organisms self-constructing Kantian Wholes do not choose to call them autopoietic unities? Don't you think this can be more evocative in explanatory terms?

3-   The authors assume that living beings know (and co-evolve) with their environment by appealing to the Gibson’s notion of affordance. However, this notion has been revised within the cognitive sciences. At least from the neurosciences and the study of the brain and consciousness, enaction instead of affordance has more empirical evidence and a solid theoretical base (Varela et el, 1991). Enaction is close tied to the explanatory framework of Maturana's biology of cognition, Friston's active inference (free energy minimization) and/or Rosen's anticipation. These ones offer causal explanations of biological cognition and perception inherent in the biological organization and realization of self-fabrication. In this sense enaction has deeper biological explanation than affordance since has its origin in biology rather than in psychology. It evokes how living systems, in their self-referential and therefore semantic organization, make sense of their environment in the way they not only use X to accomplish Y, but co-construct X to accomplish Y. Thus, the difference between enaction and affordance is not of degree, but of kind. Enaction is self-referential not only with respect to the organism’s organization, but also with the organism-environment structural coupling. That is, in enaction, neither of the two pre-exists the other, which is not the case in Gibson's affordance. In the latter, an environmental object exists independently of the organism action and sense-making. So, why do the authors prefer to choose a notion less related to biology and weaker in explanatory terms of biological cognition?

4-   Finally, there is a theorem of Robert Rosen, who some biologists think is the biology’s Newton [see Mikulecky, D. C. (2001). Robert Rosen (1934-1998): a snapshot of biology's Newton. Computers and Chemistry4(25), 317-327] that explains itself and is capable of explaining the world. This is in the mathematical language of category theory, which the great mathematician Rene Thom mentioned to be the mathematics of living and self-referential systems and therefore of semantic systems. This theorem states; a natural system is alive, if and only if, it is closed to efficient causation.This has many implications for biology in general and for the rest of science in particular, especially current physics. Having said all this, I would like to suggest to the authors that, for the sake of accuracy, the title of their paper be "The world is not a syntactic theorem."

Author Response

First of all we would like to thank to the reviewers for carefully reading the manuscript and providing very useful comments and suggestions, which helped us improving the paper and clarifying some points. We are also pleased that they have liked the paper. Moreover, we very much appreciate the observations and suggestions that have been raised, often with a view that goes beyond the ideas of our paper and that finds us in agreement.

In the following we explain how we changed the paper in response to the reviewers' requests and suggestions. Please note that for ease of the reviewers we also provide a PDF file in which the relevant parts of the text that were changed have been highlighted as colored text (added text in blue and deleted text in red). Please see attachment.

* Answer to comment 1:

We have amended the text making explicit that we refer to "syntactic" processes. However, we prefer to keep the title as it is, as we believe that the clarification in the text is sufficient.

We have also added references to mathematical approaches that have been proposed to overcome the limits of set theory. We do not discuss them in the paper, as we would rather focus on the limits of set theory.

* Answer to comment 2:

We explicitly added in the text that Kantian Wholes are examples of autopoietic systems. The concept of the Kantian Whole is that the parts exist for and by means of the whole, which allows us to define the functions of a part as that subset of the causal properties that sustains the whole. For this reason we prefer to use this expression rather than autopoietic systems. Autopoiesis by itself does not allow us to define the function of a part.

* Answer to comment 3:

We agree about the importance of enaction. However, we have on purpose limited ourselves to focus on the evolution of functional morphologies precisely to avoid issues of anything like cognition and consciousness. Our main point is that the evolution of novel functionalities precludes the use of mathematics based on set theory. A fortiori it does also with respect to behavioral aspects and we tackle these issues in a further paper currently under review available as a preprint (see ref. 6 in the revised version of the manuscript. "How organisms come to know the world: fundamental limits on artificial general intelligence", https://osf.io/yfmt3).

We also clarified the point regarding affordances and organisms in the text by remarking that an affordance can be a feature of another organism and it is reciprocal: we, as organisms, mutually create affordances between us. We have also added a further reference (see ref. 5 in the revised version of the manuscript).

* Answer to comment 4:

We do agree on this point. In fact, we have elaborated more on Rosen's work in the paper under review we mentioned above (see ref. 6 in the revised version of the manuscript) but here we just focus on set theory.

Reviewer 2 Report

The present article is one of the very few articles that are both philosophically and technically (mathematically) interesting and timely. I would thus recommend publishing this article in your journal and considering the journal lucky to have gotten this submission. I consider myself lucky to be asked to review it, so I could hopefully contribute my two cents (the authors might or might not consider any of the points below.)

The article presents a fundamental argument against the possibility of adequately describing biological evolution in terms of mathematics based on set-theory. A fortiori, if human creativity is continuous with (or a product) of biological novelty creation, it presents even an argument why the language of set-theory is the wrong language to speak about the creative process as such.

The master argument revolves around the often encountered but hard to pin down notion of “affordances”, which the authors define as “The use of X to accomplish Y” (l. 53). Such affordances have some interesting properties:

  • It is a priori unclear how and whether something contributes to the fitness of a particular organism. Flight feathers are fortuitous exaptation and originally emerged for thermoregulation, not “for flying”.
  • Affordances cannot be counted, nor ordered. How many ways are there to use an engine block? One could, of course, use it for housing an engine but also for “cracking open a coconut.” What was the “first” use of a specific protein? We might answer that after the fact (though it’s difficult!) but we cannot deduce this from first principles.

But, according to the authors, these properties violate the conditions needed to be in place to apply set theory as modeling tools! (The relevant axioms are not fulfilled).  

  1. Axiom of extensionality. Since we cannot prove whether two sets will contain the same members (affordances, i.e. “uses of Y”), this axiom is not satisfied. Note that this stems from a problem related to diachronicity: What uses can possibly be adopted in the future cannot be established at any one point in time.
  2. Axiom of choice. The axiom of choice is equivalent to sets being well-ordered (Zermelo’s well-ordering theorem), but such (diachronic?) orderings are not possible for affordances. There is no “least element” that can be identified.

This makes a strong case against trying to token affordances in terms of sets and thus derive more powerful mathematical objects based on this idea. (In particular, the idea of a “trajectory in biological phase space” does not make any sense.) The concept of numbers provides an easy example that the authors give for a more powerful mathematical tool that is built on the notion of sets. Natural numbers have traditionally been axiomatized by relying on a “Null-set” and a “successor-element”. Whether there is such a Null-set is not provable (l.192: “In our case this corresponds to ‘the set of all objects that have exactly 0 uses.’ Well, No.”) Neither is the claim that anything like a “successor element” exists (l. 195: “the different uses of X are unordered”).

The rest of the article deals with rebutting certain objections, most having to do with a misunderstanding of the idea that we simply do not know what could be relevant or not for biological evolution.  

While one may be tempted to gloss over the notion of “set-theory”, it is important to note that this does not, by itself, rule out the applicability of mathematics per se, but the applicability of mathematics based on set-theory. Although the philosophical details are still disputed, many working mathematicians would not doubt that set-theory is the most fundamental theoretical enterprise that grounds contemporary mathematics. So doubting that set-theory is adequate really poses a problem! (As an aside, almost no mathematician works on “set-theory” anymore, those who do are mostly philosophers of mathematics. Be that as it may, the articles’ focus on set-theory is great since it allows one to shake up the fundamentals w/o going into the nitty-gritty details, e.g. not having to learn topology or category theory first.)

I have previously proposed a mathematical model of how to construct “phenomenal spaces” (topological spaces representing complex experiences) based not on sets of point-like entities but based on collections of extended elements. I speculated that affordances might encode the basic (non-point-like, relational, and processual) elements out of which one can construct a topological space, given (i) that one has a notion of “sameness” (which is however dependent on having a memory that lets one say “same” or “not same”) and (ii) that one can stipulate boundaries that distinguish “collection X” from “collection Y” (which seems to me to be irreducible to a simple mechanical procedure: what boundaries are at all possible is not pre-stateable). I wonder whether this is part of what the authors conjecture at the end when they say “Perhaps new kinds of mathematics can be invented” (l. 319).

Robert Prentner. “Consciousness and topologically structured phenomenal spaces.” Consciousness & Cognition 70, 25-39, 2019.

--

Minor comments and questions:

  • I would be curious to know to what extent the authors believe the problems outlined are “epistemic” or “ontological” obstacles? E.g. one might perhaps assume that 3N degrees of freedom (where N is the number of molecules of the organism) specify, in principle, a viable (physical) phase space but would be completely unwieldy to do any serious calculation – hence an “epistemological” problem. The discussion of the “Laplace demon” in the objection seems to suggest such a limitation related to knowledge (“we cannot even know the functional items that are relevant for our predictions”) Alternatively, one might think that this problem is more “ontological” in nature (hence serious). g. phase spaces encode the relevant invariants of motion (symmetries). But it is not clear that there are any such things in biology...? I’m a bit puzzled by this. This question is also related to non-reductionism: epistemological or ontological
  • Typo on l. 141 “One might argue that if we had a computational for the evolution of the biosphere,” is missing something after computational, e.g. “computational model”?
  • The discussion of “ergodicity” at the beginning of section 3 would perhaps better fit in the objections section. Similar for the (interesting) aspect of self-construction: Often, affordances are used to accomplish self-maintenance or self-construction, i.e. a cell “uses” the contingently available chemicals and means to manipulate them to construct itself. I have a hunch this implied circularity poses a big problem for computability issues, but I am not exactly sure how to spell this out (e.g. similar claims about non-Turing computability due to maintenance have been made by Robert Rosen in the 80ies). Both "ergodicity" and "self-constructing" are interesting and important characteristics of biological systems, but I don't see the connection to the argument against set-theory.

Author Response

First of all we would like to thank to the reviewers for carefully reading the manuscript and providing very useful comments and suggestions, which helped us improving the paper and clarifying some points. We are also pleased that they have liked the paper. Moreover, we very much appreciate the observations and suggestions that have been raised, often with a view that goes beyond the ideas of our paper and that finds us in agreement.

In the following we explain how we changed the paper in response to the reviewers' requests and suggestions. Please note that for ease of the reviewers we also provide a PDF file in which the relevant parts of the text that were changed have been highlighted as colored text (added text in blue and deleted text in red). Please see the attachment.

* Answer to comment on new mathematics:

We thank the reviewer for point us out a very interesting work on a mathematical model on the construction of "phenomenal spaces". We have added a citation to this work.

* Answer to question about ontological vs epistemic

This is a very profound question. We think the answer is ontological, because possibilities are ontological. The reviewer may find our considerations on this subject in a paper that will be submitted shortly (available as a preprint "What Is Consciousness? Artificial Intelligence, Real Intelligence, Quantum Mind, and Qualia", https://arxiv.org/abs/2106.15515)

* Answer to question about typo:

Corrected. Thanks for spotting it.

* Answer to question about ergodicity:

We prefer to keep the term "ergodicity" in Section 3, since non-ergodicity is required for introducing the notion of Kantian Wholes and it allows us also to define "functions", which cannot be defined non-circularly. It is exactly because of these reasons that we cannot use set theory for modeling the evolution of new organisms and functions.
